# Effect of Corn Straw on Hydrogen Production from Lignite

Ying Wang [1], Litong Ma [1,2,3,*] and Jun Li [4]

1. School of Chemistry and Chemical Engineering, Inner Mongolia University of Science and Technology, Baotou 014010, China
2. Inner Mongolia Engineering Research Center of Comprehensive Utilization of Bio-Coal Chemical Industry, Baotou 014010, China
3. Laboratory of Low Rank Coal Carbon Neutralization, Inner Mongolia University of Science and Technology, Baotou 014010, China
4. School of Life Science and Technology, Inner Mongolia University of Science and Technology, Baotou 014010, China
* Correspondence: mlt0916@126.com

**Abstract:** The conversion of lignite to clean energy has won considerable attention and plays an important role in achieving the goal of carbon reduction. The effects of corn straw on hydrogen production from lignite was explored by using lignite as the substrate and corn straw as an exogenous substance. The fermentation mechanism was elucidated through the analysis of total and daily hydrogen production; the concentration of humic acid, benzoic acid, pyruvate, and glucose, as well as pH value. In addition, total solid (TS), and volatile solid (VS) from activated sludge before and after fermentation are examined. The results showed that corn straw could accelerate hydrogen production from lignite with an optimal content of corn straw of 40%. The fermentative hydrogen production with 40% corn straw was up to 186.20 mL, 3.40 times higher than that of the control group. Corn straw effectively improved the concentration of humic acid and benzoic acid, accelerating the anaerobic fermentation of lignite to produce hydrogen. The concentration of pyruvic acid, glucose, pH, and the changes in TS and VS before and after fermentation showed that the group of 40% corn straw had a better promotion effect than other systems for hydrogen production. This provides a new idea for improving hydrogen production through lignite anaerobic fermentation.

**Keywords:** lignite; hydrogen; humic acid; benzoic acid; pyruvic acid

## 1. Introduction

China is rich with lignite, but it has high moisture content, low calorific value, and low combustion efficiency [1]. During the lignite burning, a large amount of smoke is produced, which causes serious environmental pollution. It was found that lignite can be converted into clean energy to reduce environmental pollution [2–5]. At the same time, hydrogen is more advantageous than any fuel because it is the cleanest energy [6]. At present, lignite can produce hydrogen by electrolysis, gasification, and anaerobic fermentation. The electrolytic cell designed by Luy et al. enables Faradaic efficiency for hydrogen evolution to reach 99% [7]. However, the electrolysis of hydrogen requires costly investments to be performed, thus, in practical applications, it is not economical. Yu et al. and He et al. [8,9] both increased hydrogen production by adding catalysts to lignite gasification systems. However, there are high-temperature and high-pressure conditions in the lignite gasification process. With a low cost and simple process, anaerobic fermentation is widely used to obtain hydrogen and has become one of the most promising technologies [10]. Nonetheless, there is a common problem of low hydrogen yield in fermentative hydrogen production from lignite. In order to improve hydrogen production, many options were explored. Xia et al. [11] reported that acid-pretreatment and alkali-pretreatment of lignite could enhance the production of hydrogen, and alkali-pretreatment produced more hydrogen. Chen et al. [12] found that the addition of $Fe^{3+}$ and $Ni^{2+}$ could effectively promote the lignite to produce hydrogen,

because $Fe^{3+}$ and $Ni^{2+}$ are an important part of ferredoxin, iron sulfur protein, urease, etc. If sugar-rich substances were added to improve microbial activity, lignite could produce more hydrogen.

Corn is one of the most widely planted crops in China [13]. However, corn straw is usually used for animal silage and burning. Actually, a large amount of cellulose and hemicellulose in corn straw can be used for fermentative hydrogen production [14]. He et al. [15] found that basalt fiber increased the hydrogen production of corn straw because basalt fiber improved microbial activity. Furthermore, Chen et al. [16] found that acid pretreatment of corn straw increased hydrogen production by promoting cellulose hydrolysis. At present, the co-fermentation of lignite and straw is also being studied. Yoon et al. [17] found that the addition of substrate straw (6.25% $w/w$) inhibited methane production, and with the increase in straw, methane production showed an increasing trend. Guo et al. [18] studied the effect of the fermentation of lignite and corn straw on methane production, and the results showed that the optimal ratio of lignite and corn straw to biomethane was 2:1. However, there is no research concerning the effect of adding corn straw on hydrogen production in lignite.

In this paper, hydrogen production from anaerobic fermentation of lignite with corn straw was studied. Moreover, changes in total hydrogen production, daily hydrogen production, humic acid, benzoic acid, and pyruvic acid concentration, as well as pH, were studied. In addition, the changes in TS and VS from activated sludge before and after fermentation were explored.

## 2. Materials and Experimental Methods

### 2.1. Materials

In this study, the lignite was collected from Pingzhuang Coal Mine, Chifeng City, Inner Mongolia Autonomous Region. Corn straw was taken from Machi Town, Baotou City, Inner Mongolia Autonomous Region. Anaerobic activated sludge was obtained from the sewage treatment plant in Baotou. Tables 1 and 2 show parameters of lignite.

**Table 1.** Lignite industry analysis.

| Moisture-Total ($M_t$)/% | Ash ($A_d$)/% | Volatile Compound ($V_d$)/% | Fixed Carbon ($F_c$)/% |
|---|---|---|---|
| 6.44 | 15.86 | 34.13 | 43.57 |

**Table 2.** Analysis of physical and chemical properties of lignite.

| Material | Organic Matter | Total Humic Acid | Water-Soluble Humic Acid |
|---|---|---|---|
| lignite | 51.48 | 24.60 | 5.94 |

### 2.2. Experimental Methods

Lignite anaerobic fermentation was performed using an automatic methane potential test system (AMPTS II, Bioprocess, Lund, Sweden) in a 500 mL fermentation flask. As mentioned in Table 3, each bottle was fed with lignite and corn straw crushed to 100 mesh and 200 mL of anaerobic activated sludge. Parameters of TS and VS of activated sludge are shown in Table 4. The total volume was set to 400 mL with distilled water, and the pH was adjusted to 7.0. Bottles were placed in a constant temperature water bath at 50 °C. The stirring was automatically controlled by the motor and occurred every 2 h for 10 min each time. The daily hydrogen production data were regularly recorded every day. The total hydrogen production was calculated after the experiment. The fermentation solution was taken every 3 days to measure the concentration of humic acid, benzoic acid, pyruvate, and glucose and pH in the fermentation process. The TS and VS of the activated sludge were measured before and after fermentation. All the experiments were performed

independently in triplicate. We collected 5 mL fermentation broth at regular intervals, followed by centrifugation at 10,000 rpm for 10 min. Then, the supernatants were analyzed.

**Table 3.** Different contents of corn straw added in the fermentation.

| Test Group | Lignite/g | Corn Straw/g |
|------------|-----------|--------------|
| A0 | 40 | 0 |
| A1 | 36 | 4 |
| A2 | 32 | 8 |
| A3 | 28 | 12 |
| A4 | 24 | 16 |
| A5 | 20 | 20 |

**Table 4.** The parameters of activated sludge.

| Parameters | Anaerobic Activated Sludge |
|------------|----------------------------|
| TS | 0.23% |
| VS | 0.11% |
| VS/TS | 0.48 |

*2.3. Chemical Analysis*

The TS content was determined by drying activated sludge, in triplicate, at 105 °C for 12 h in the muffle furnace. The VS content was measured after burning samples at 550 °C for 2 h. After the supernatants were diluted, humic acid was determined by ultraviolet detection at 256–400 nm and the absorbance was averaged [19]. Benzoic acid concentration was measured using high-performance liquid chromatography (Agilent-1260) with an ultraviolet detector and C18 analytical column (Poroshell, 4 μm, 4.6 × 150 mm). The mobile phase contained ammonium acetate (0.02 mol/L): methanol = 92:8 (*v/v*) with a flow rate of 1 mL/min. The concentration peak of benzoic acid was measured at 230 nm wavelength. The injection volume was 20 μL. Pyruvate concentration was determined by ultraviolet detection [20]. The supernatants were diluted and adjusted to pH = 11 for determination at 320 nm. The glucose concentration was measured using Agilent-1260 high-performance liquid chromatography with Hi-Plex H column. The mobile phase was 5 mM $H_2SO_4$, and the flow rate was 0.6 mL/min. The detector was a difference detector. The column temperature and detector temperature were 60 °C and 55 °C, respectively [21]. The pH of the fermented broth was measured with a digital pH-meter (LEICI Instruments pHS-3C, Shanghai, China).

## 3. Results and Discussion

*3.1. Change in Total Hydrogen Production with Different Contents of Corn Straw in Fermentative Hydrogen Production from Lignite*

The total hydrogen production achieved with different contents of corn straw was plotted in Figure 1. The total hydrogen production was 54.73, 67.30, 83.45, 148.37, 186.20, and 123.35 mL with 0%, 10%, 20%, 30%, 40%, and 50% corn straw, respectively. These values were 22.97%, 52.48%, 171.09%, 240.22%, and 125.38% greater than those in the control group without corn straw, respectively. The results showed that corn straw could increase the total hydrogen production from lignite. The group with 40% corn straw showed the highest total hydrogen production and was the most promising. The microbial activity was improved because corn straw is rich in nitrogen [22]. Chen et al. found that the addition of $Fe^{3+}$ and $Ni^{2+}$ also increased hydrogen production by increasing microbial activity [12]. However, the effect of adding 50% corn straw in hydrogen production from lignite was lower than that of adding 40% corn straw. This was because in the fermentation process of the A5 group, more $CO_2$ was generated, impairing the microbial activity in the system [5].

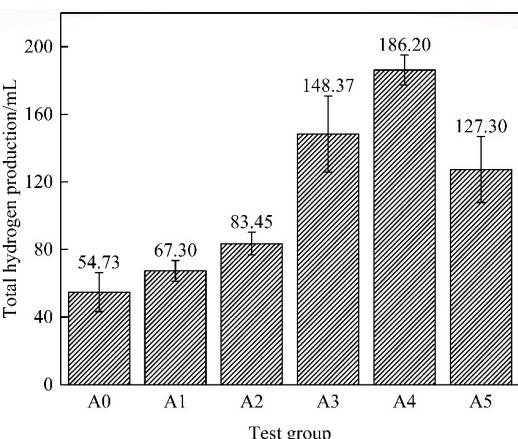

**Figure 1.** Change in total hydrogen production with different contents of corn straw in fermentative hydrogen production from lignite.

### 3.2. Change in Daily Hydrogen Production with Different Contents of Corn Straw in Fermentative Hydrogen Production from Lignite

Adding corn straw had obvious effects on hydrogen production from lignite (Figure 2). The control group and experimental A1, A2, A3, A4, and A5 groups achieved maximum daily hydrogen production at the first day and then, tended to zero. The peak of daily hydrogen production from the control group was 5.37 mL. In addition, the peaks of the A1, A2, A3, A4, and A5 groups were 18.13, 36.37, 61.35, 93.67, and 19.67 mL, respectively. The fermentation in the groups with the addition of corn straw was higher than that of the control group. Adding corn straw improved the hydrogen production rate of lignite. On the first day, the hydrogen production of the A4 group was 17.44 times higher than that of the control group. Therefore, adding 40% corn straw significantly enhanced hydrogen production from lignite. Although the daily hydrogen production of the A5 group was higher than that of the A4 group in the later stage of fermentation, the peak hydrogen production of A4 group was much higher than that of the A5 group. This showed that adding 40% corn straw had the most obvious enhancing effect on hydrogen production from lignite, because corn straw is rich in cellulose, which could be hydrolyzed to glucose and then provide nutrients for microorganisms [23]. At the same time, humic acid can be degraded in the fermentation process of corn straw, and a small amount of humic acid can improve the activity of microorganisms and, thus, increase hydrogen production [24,25]. This is consistent with the research results of Wang et al., which showed that adding a small amount of water-soluble fulvic acid in humic acid into the fermentation system improved microbial activity [26].

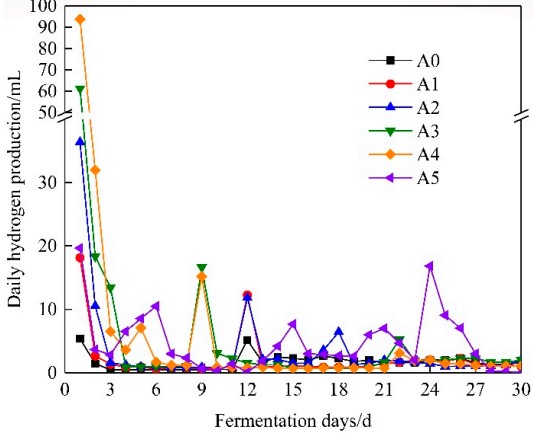

**Figure 2.** Change in daily hydrogen production with different content of corn straw in fermentative hydrogen production from lignite.

### 3.3. Change in Humic Acid Concentration with Different Contents of Corn Straw in Fermentative Hydrogen Production from Lignite

The humic acid concentration achieved with different contents of corn straw is shown in Figure 3. Because a large amount of humic acid was consumed and hydrogen was produced in the experimental groups on the first day, the concentration of humic acid was lower at the beginning of fermentation. The concentration of humic acid in the control group and experimental A1, A2, A3, A4, and A5 groups was 1.34, 2.89, 3.04, 0.33, 0.51, and 4.04 g/L, respectively. This was consistent with the A3 and A4 groups producing more hydrogen. At the same time, when 50% corn straw was added the concentration of humic acid was higher, but the daily gas output was lower because the high concentration of humic acid inhibited the activity of microorganisms and reduced the efficiency of conversion to hydrogen [27]. This is consistent with a study by Wang et al., wherein excessive water-soluble humic acid (fulvic acid) inhibited microbial activity in the fermentation system [28]. Then, the hydrogen production decreased and humic acid concentration increased. On the 3rd day, the concentration of humic acid from the control group and experimental A1, A2, A3, A4, and A5 groups was 3.04, 4.22, 4.23, 4.74, 5.14, and 4.60 g/L, respectively. The addition of corn straw increased the concentration of humic acid because the lignin in corn straw could be decomposed into phenolic compounds and then converted into humic acid [29,30]. Afterwards, humic acid was converted to benzoic acid, reducing the concentration during the fermentation process. On the 12th day, humic acid concentration decreased to 0.94, 0.63, 0.56, 0.71, 0.65, and 0.87 g/L with respect to the control group and A1, A2, A3, and A4 groups. This represented a decrease of 69.04%, 85.06%, 86.85%, 85.02%, 87.27%, and 81.09%, respectively, with respect to the 3rd day. This suggested that the biotransformation ability was stronger and more humic acid could be converted to hydrogen by adding 40% corn straw to the fermentation system, because humic acid was also promoting a microbial-mediated electron-transfer chain reaction [25,27,31].

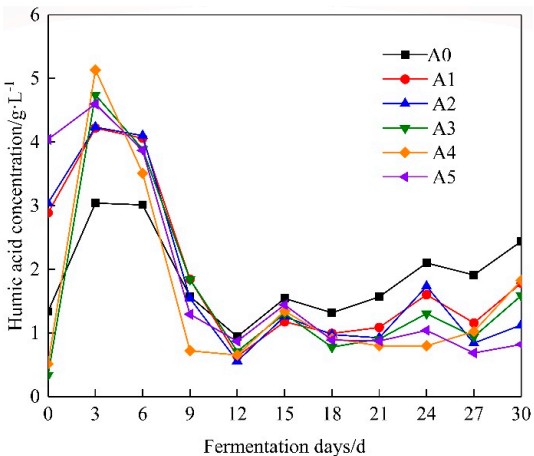

**Figure 3.** Change in humic acid concentration with different contents of corn straw in fermentative hydrogen production from lignite.

### 3.4. Change in Benzoic Acid Concentration with Different Contents of Corn Straw in Fermentative Hydrogen Production from Lignite

Figure 4 shows the benzoic acid concentration in different groups. There was no significant fluctuation in the concentration of benzoic acid from the control group and the A1 group during the fermentation process. The control group and the A1 group reached the maximum concentration of benzoic acid on the 3rd day, with 1.16 and 1.50 mg/L, respectively. However, the concentration of benzoic acid in groups A2, A3, A4, and A5 showed a trend of increase and then decrease. In addition, they all reached the concentration peak on different days, with 2.71, 7.52, 11.35, and 4.74 mg/L, respectively. This was consistent with the extent of humic acid degradation, because humic acid can be degraded to benzoic acid [32]. The group A4 showed that the concentration peak increased by

878.45% compared with that of the control group. Therefore, there was more benzoic acid to be converted to pyruvic acid, and then, hydrogen was produced [33]. This indicated that adding corn straw has significant enhancing effects on hydrogen productivity. Likewise, the addition of 40% corn straw had the most obvious enhancing effect on hydrogen production from lignite.

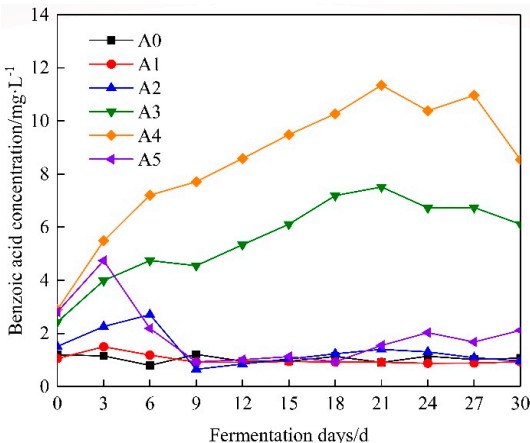

**Figure 4.** Change in benzoic acid concentration with different contents of corn straw in fermentative hydrogen production from lignite.

*3.5. Change in Pyruvic Acid Concentration with Different Contents of Corn Straw in Fermentative Hydrogen Production from Lignite*

The effect of adding corn straw on pyruvic acid is shown in Figure 5. Similar trends were observed in different fermentation groups. Figure 5 shows that during fermentative hydrogen production from lignite, the pyruvic acid increase fluctuated but fell to a minimum on day 27. The minimum concentrations of pyruvic acid in the control group and experimental groups A1, A2, A3, A4, and A5 groups were 29.96, 13.98, 14.88, 12.62, 11.98, and 13.07 g/L, respectively. The pyruvic acid concentration reached the minimum with the corn straw content of 40%. On the 27th day, the pyruvic acid concentration of the control and A4 groups was 6.50% and 65.78% lower than that of the 3rd day, respectively. Pyruvic acid initiates decarboxylation to produce acetyl CoA and then hydrogen generation [34]. Therefore, the higher degradation degree of pyruvic acid in group A4 indicated that the enhancement of hydrogen production in group A4 was clear.

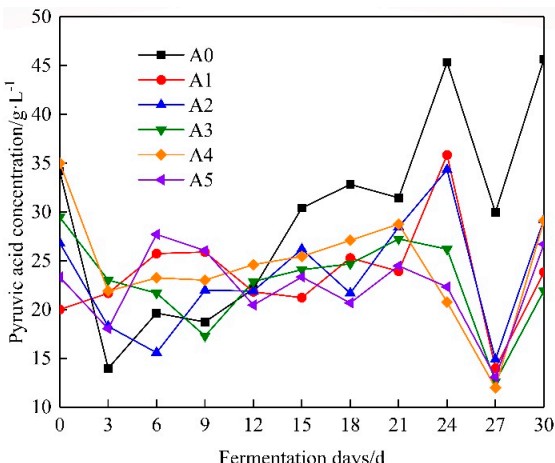

**Figure 5.** Change in pyruvic acid concentration with different contents of corn straw in fermentative hydrogen production from lignite.

### 3.6. Change in Glucose Concentration with Different Contents of Corn Straw in Fermentative Hydrogen Production from Lignite

Figure 6 shows that the trends in glucose concentration in different fermentation groups were different. The minimum glucose concentration of the control group and experimental A1, A2, A3, A4, and A5 groups were 3.98, 6.05, 6.37, 7.57, 6.03, and 14.40 mg/L, respectively. The more corn straw was added, the higher the glucose content was, because corn straw is rich in cellulose that can be rapidly degraded into glucose [35]. The water solubility of lignite was poor, and it was difficult to break the bonds to produce glucose. However, the lowest glucose concentration in group A4 was even lower than that in group A3, indicating that 40% corn straw could enhance the microbiological activity [36], thus, promoting the degradation of glucose and generating more hydrogen. However, the excessive corn straw reduced the glucose degradation, as a large amount of the carbon source dissolved from corn straw made fermentation change to an acid production stage [37]. Large amounts of acid reduced microbial activity, making glucose difficult to degrade, inhibiting hydrogen production.

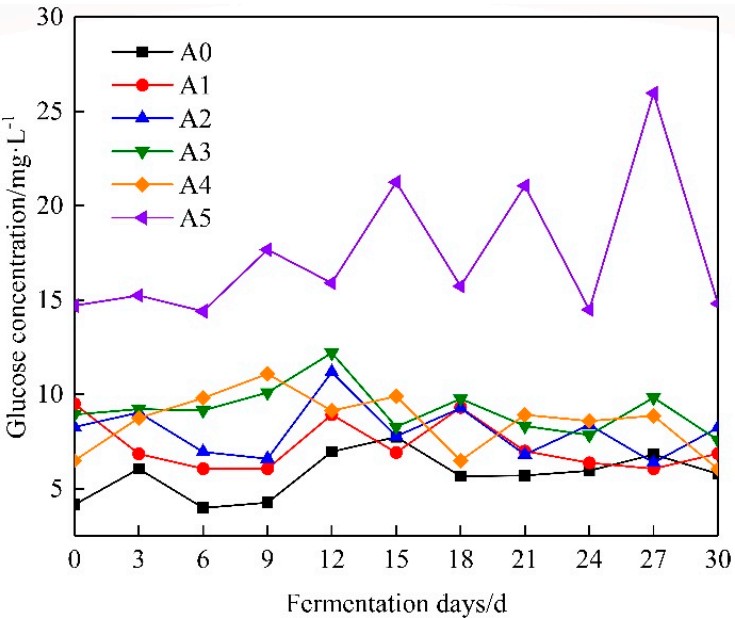

**Figure 6.** Change in glucose concentration with different contents of corn straw in fermentative hydrogen production from lignite.

### 3.7. Change in pH with Different Contents of Corn Straw in Fermentative Hydrogen Production from Lignite

In the process of microbial fermentation, the pH was one of the main factors affecting microbial activity, which in turn, affects the hydrogen production and the stability of the fermentation system. The pH change in different lignite fermentation groups is shown in Figure 7. Because corn straw could produce acid substances, such as acetic acid, during fermentation, the pH value of groups with corn straw is low. On the 3rd day, the pH of all the fermentation groups decreased due to the production of acid substances such as humic acid. The pH of the control group and experimental A1, A2, A3, A4, and A5 groups was 6.51, 5.27, 4.93, 4.86, 4.67, and 4.91, respectively. The A4 group exhibited the smallest value of pH, which indicated that more acidic substances were converted into hydrogen in the early stage of fermentation. This is consistent with the results of Xia et al., who found that the more acid content, the stronger the hydrogen production capacity [10]. Afterwards, the acid substance was consumed and the pH value increased.

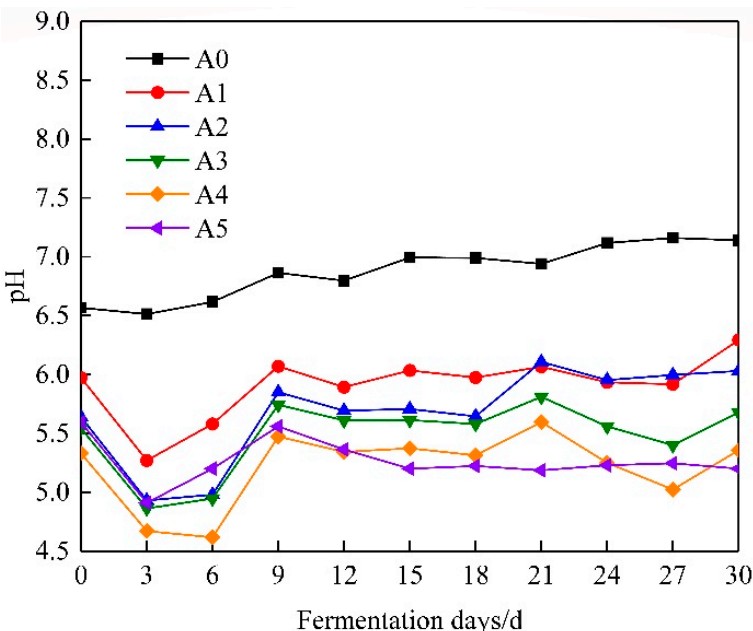

**Figure 7.** Change in pH with different contents of corn straw in fermentative hydrogen production from lignite.

### 3.8. TS and VS of Fermentation Liquid in Fermentative Hydrogen Production from Lignite

Figure 8 shows that the TS and VS contents of activated sludge were 0.15% and 0.08%. The increase in TS after domestication may be caused by the increase in suspended solids and the microbial reproduction number with corn straw and lignite. After domestication, because organic matter content increased in soluble substances, the VS content increased [38]. With this, the organic matter content was higher and the hydrogen fermentation ability stronger. With the increase in exogenous corn straw contents, the TS and VS of the fermentation liquid increased and then decreased. TS and VS reached the maximum at a corn straw content of 40%, which increased by 112.82% and 52.17% compared to the control group. Therefore, adding 40% corn straw is better to enhance hydrogen production.

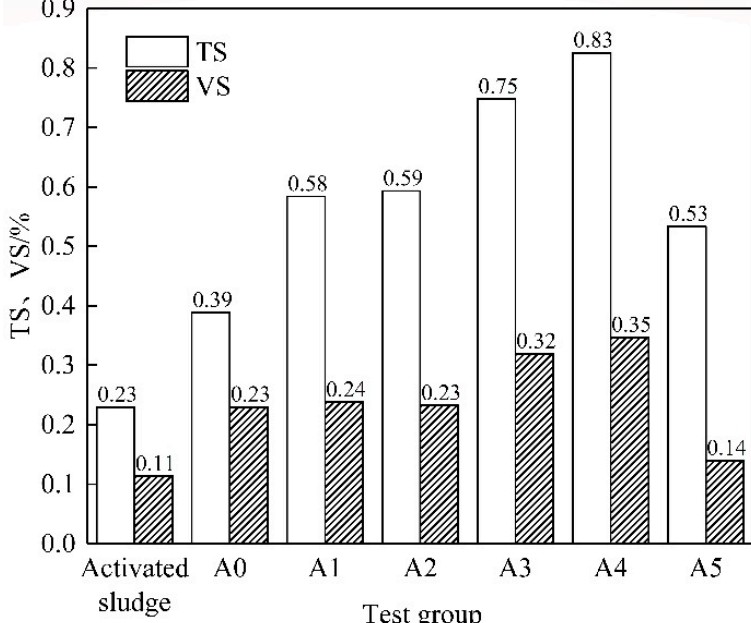

**Figure 8.** TS and VS of fermentation liquid in fermentative hydrogen production from lignite.

### 4. Conclusion

(1) The total hydrogen production in the lignite fermentation groups enriched with corn straw was higher than that of the control group. The total hydrogen production in the lignite fermentation system enriched with 40% corn straw was 186.20 mL, 2.40 times higher than that of the control group. This is because the addition of corn straw can increase the nitrogen content, thus, regulating the C/N ratio of the fermentation system. At the same time, the cellulose in corn straw can be degraded into glucose, improving the activity of microorganisms.

(2) The addition of 40% corn straw can increase the concentration of benzoic acid, humic acid, pyruvic acid, and glucose in the fermentation liquid and promote their degradation. This is because the humic acid and glucose in the fermentation liquid also improve the activity of microorganisms.

(3) The contents of TS and VS in the fermentation liquid with 40% corn straw fermentation group were the highest, indicating that activated sludge contained more organic matter and produced more hydrogen.

**Author Contributions:** Data curation, and writing—original draft, Y.W.; conceptualization, formal analysis, funding acquisition, methodology, and writing—review and editing, L.M.; writing—review and editing, J.L. All authors have read and agreed to the published version of the manuscript.

**Funding:** This work is financially supported by Chinese Academy of Sciences "Light of the West" Program (2019); Science and Technology Plan Project of Inner Mongolia Autonomous Region (2020GG0158); Talent Development Fund Project of Inner Mongolia Autonomous Region (2021); Fundamental Research Funds for Inner Mongolia University of Science and Technology (2022).

**Conflicts of Interest:** The authors confirm that this article's content contains no conflict of interest.

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
