# Peer review of "Effect of Corn Straw on Hydrogen Production from Lignite"

_fermentation, doi:10.3390/fermentation9020106_

Round 1

Reviewer 1 Report

Effect of Corn Straw on Hydrogen Production from Lignite

Authors developed the process for hydrogen production from lignite and co-substrate of corn straw. The fermentation mechanism had been understood through the analysis of total hydrogen production and others. The results showed that corn straw could accelerate hydrogen production from lignite. The results are interesting to the related research fields. The work has good novelty and should be considered for publication after some revision.

1 The language of this manuscript should be significantly improved

2 Abbreviation in the text should meet the requirements of the journal

3 “China is rich in lignite, but it is high moisture content, …” grammar error

4 Introduction should include the progress of co-utilization of biomass and lignite

5  “Table 3 Different concentrations of corn straw added” should be “Table 3 Different contents of corn straw added in the fermentation”

6 The detailed information of fermentation should be provided in the methods section

7 Error bar should be provided in figure 1

8 More discussion about the results are needed by comparing with other reports

9 Conclusion section should be rewritten, and it is suggested to provide the key results and main conclusions

Reviewer 2 Report

Paper describes one of the most natural mechanisms of the hydrogen production from the organic matter - the fermentation mechanism. The lignite-glucose composition shows the prominent perspective for production of the green hydrogen. However, there are some unclear points that have to be clarified:

1. According to Fig.2, the sample A5 produces the increasing pulses of hydrogen production with time.  Did you check the process efficiency for a longer time? May be the A5 will be more effective for a durable coversion process because the level of humic acid for A5 is lowest after 27 days.

2. The trend of the glucose content in Fig.6 for sample A5 is growing in average. What does it mean in context of the fermentation process which consumes the glucose as I indestand?

3. As authors declare, the pH of composition reflects the intensity of bio-conversion. Looking onto the Fig.7, most of the samples show the evident minimum at 3 day and the A4 has one more at 27 day.  
However, the A5 has longterm ranges with low level of pH (lower than others). Can it reflect the smaller but durable acids production that means the possibility of other effective  compositions?

Author Response

  1. According to Fig.2, the sample A5 produces the increasing pulses of hydrogen production with time. Did you check the process efficiency for a longer time? May be the A5 will be more effective for a durable conversion process because the level of humic acid for A5 is lowest after 27 days.

Response: Thanks for your good comments. As shown in Figure 2, hydrogen production of control group and experimental A1, A2, A3, A4, and A5 groups reached the concentration peak on the first day, and then fluctuated slightly and eventually tended to 0. When there was no daily hydrogen production for three consecutive days, we finished the fermentation. The hydrogen production of A5 group increased with the time, but it still dropped to zero eventually. Therefore, the process efficiency was not farther checked. In addition, the role of humic acid in microbial degradation and the transformation of aromatic compounds is still controversial, and its role in biomethanation is also unclear. The effect of humic acid on hydrogen fermentation of lignite is still being studied.

  1. The trend of the glucose content in Fig.6 for sample A5 is growing in average. What does it mean in context of the fermentation process which consumes the glucose as I understand?

Response: Thanks for your good comments. When corn straw is added to the reaction system to produce hydrogen from lignite, the cellulose degradation of corn straw and the humic acid degradation of lignite produce glucose.  At the same time, glucose is converted into hydrogen by microorganisms and consumed.  The glucose produced exceeds the glucose consumed, leading to an average increase in the glucose content of the reaction system.

  1. As authors declare, the pH of composition reflects the intensity of bio-conversion. Looking onto the Fig.7, most of the samples show the evident minimum at 3 day and the A4 has one more at 27 day. However, the A5 has long-term ranges with low level of pH (lower than others). Can it reflect the smaller but durable acids production that means the possibility of other effective compositions?

Response: Thanks for your good comments. Corn straw is added to the reaction system of lignite to produce hydrogen. A5 means that 50% corn straw is added to the reaction system, indicating the amount of corn straw is the most. The six-carbon sugar (glucose) produced by cellulose degradation of corn straw, along with the five-carbon sugar produced by hemicellulose degradation, will be degraded by microorganisms and produce volatile fatty acids such as formic acid, acetic acid, propionic acid, and butyric acid.   The conversion rate of active components such as formic acid, acetic acid, propionic acid, and butyric acid is different, which will accumulate in the reaction system within a certain period, resulting in the low pH range of the A5 group for a long time.